# A Pharmacovigilance Study on the Safety of Axicabtagene Ciloleucel Based on Spontaneous Reports from the EudraVigilance Database

**DOI:** 10.3390/biomedicines11082162

**Published:** 2023-08-01

**Authors:** Concetta Rafaniello, Valerio Liguori, Alessia Zinzi, Mario Gaio, Angela Falco, Luigi Di Costanzo, Francesca Gargano, Valentina Trimarco, Mauro Cataldi, Annalisa Capuano

**Affiliations:** 1Campania Regional Centre for Pharmacovigilance and Pharmacoepidemiology, 80138 Naples, Italy; concetta.rafaniello@unicampania.it (C.R.); valerio.liguori@unicampania.it (V.L.); alessia.zinzi@unicampania.it (A.Z.); mario.gaio@unicampania.it (M.G.); angelafalco992@libero.it (A.F.); 2Section of Pharmacology “L. Donatelli”, Department of Experimental Medicine, University of Campania “Luigi Vanvitelli”, 80138 Naples, Italy; luisdico86@gmail.com; 3Section of Pharmacology, Department of Neuroscience, Reproductive Sciences and Dentistry, Federico II University of Naples, Via Sergio Pansini 5, 80131 Naples, Italy; valentina.trimarco@unina.it; 4Department of Anesthesia and Resuscitation, Biomedical Campus University of Rome, 00128 Rome, Italy; f.gargano@policlinicocampus.it

**Keywords:** CAR-T-cells, axi-cel, immune-effector-cell-associated neurotoxicity syndrome, cytokine release syndrome, pharmacovigilance, EudraVigilance

## Abstract

During pre-approval clinical trials, the safety of axi-cel, a second-generation CAR-T-cell therapy directed against CD19, which dramatically improved the prognosis of intractable B-cell lymphomas, has been investigated only in about 400 patients. Therefore, additional information on this issue is urgently needed. In the present paper, we evaluated the 2905 ICSRs with axi-cel as the suspected drug that had been uploaded in the EudraVigilance database from 1 January 2018 to 31 December 2022. About 80% of the reported adverse events were serious, and about 20% of them did not fully resolve or caused death. The adverse events most-frequently reported were Nervous system disorders (25.6%) and, among them, immune-effector-cell-associated neurotoxicity syndrome, followed by Immune system disorders (23.1%), General disorders and administration site conditions (12.0%), Blood and lymphatic system disorders (7.2%), and Infections and infestations (5.8%). Disproportionality analysis showed that the frequency of reported adverse events related to the nervous system was higher with axi-cel than with the other approved CAR-T-cells, except brexu-cel. In conclusion, real-world pharmacovigilance data showed that nervous system and immune system disorders are the adverse events most reported in axi-cel-related ICSRs and suggest that axi-cel could be more neurotoxic than other CAR-T-cells.

## 1. Introduction

Cell therapy is the administration of living cells to treat human diseases. After years of experimentation, this groundbreaking therapeutic approach is entering into clinical practice with the authorization for use in humas of several of these products [1]. Among them, Chimeric Antigen Receptor-T (CAR-T) cells have a very special relevance. CAR-T-cells are T lymphocytes collected from patients and genetically engineered in vitro to express modified T-cell receptors, which recognize specific antigens so that, when reinfused in the patient, they may trigger immune aggression against cancer cells [2]. Additional modifications have been introduced to the T-cell receptor to enhance its ability to induce an immune response against cancer cells including one (in second-generation CAR-Ts) or multiple (in third-generation CAR-Ts) costimulatory domains. Fourth-generation CAR-T-cells, also called T-cell Redirected for Universal Cytokine-mediated Killing (TRUCK) cells, besides a modified T-cell receptor bearing a single costimulatory domain, also express a second inducible transgene coding a cytokine with killing abilities against cancer cells [3].

CAR-T-cell therapy products profoundly differ from conventional drugs in several respects including their production, mechanism of action, and administration modalities, all of which also raise specific and unprecedented issues, such as the collection, safe transport, and storage of autologous cells, their reinfusion after ex vivo manipulation, and the transfer of genetically modified cells to the patient [4]. The diversity of cell therapy products form conventional drugs has been acknowledged by regulatory agencies, and the European Medicine Agency (EMA) classifies these innovative therapeutic agents in the broader class of Advanced Therapy Medicinal Products (ATMPs), which also includes gene therapy and cell engineering products. A specific regulatory framework has been implemented for ATMPs in the European Union (EU) [5], and the EMA also established a new technical committee, the Committee for Advanced Therapies (CAT), to support the Committee for Medicinal Products for Human Use (CHMP) in the evaluation of these products [6]. Considering the novelty of these products and the urgent need for further efficacy and safety clinical data, ATMPs are routinely approved under specific Risk Minimization Programs (RMPs), which are also intended to collect missing information on their tolerability through active pharmacovigilance surveillance [6]. 

Six CAR-T-cell products have been authorized by the EMA so far. Four of them (tisagenlecleucel (isa-cel), axicabtagene ciloleucel (axi-cel), lisocabtagene maraleucel (lisa-cel), and brexucabtagene autoleucel (brexu-cel)) are directed against the Cluster of Differentiation 19 (CD19) antigen, a plasma membrane integral protein selectively expressed on B-cells’ surface, which controls B-lymphocytes’ differentiation and survival and, therefore, is an ideal target for the immunotherapy of B-cell lymphomas [7]. In contrast, the remaining two approved CAR-T-cells, ciltacabtagene autoleucel (cilta-cel) and idecabtagene vicleucel (ide-cel), have been approved for refractory multiple myelomas since they target the B-Cell Maturation Antigen (BCMA), also known as Tumor Necrosis Factor Receptor Superfamily member 17 (TNFRSF17), which is expressed in mature B-cells and overactivated in this hematological neoplasm [8]. 

In the present manuscript, we focused on axi-cel, a second-generation CAR-T-cell engineered to express a chimeric antigen receptor including an anti-CD19 domain, a CD3ζ activation domain, and a CD28 costimulatory domain [9] (Figure 1). 

Axi-cel was initially approved by the Food and Drug Administration (FDA) and the EMA for relapsed or refractory Diffuse Large B-Cell Lymphoma (DLBCL) and Primary Mediastinal large B-Cell Lymphoma (PMBCL) after two or more lines of systemic therapy. This approval was based on the results of the phase 1/2 clinical trial ZUMA-1, which enrolled a total of 7 patients in the phase 1 pivotal study [10] and 101 patients in the phase-2 study [11]. More recently, after the positive results, respectively, of the phase 2 clinical trial ZUMA-5, which evaluated 109 patients [12], and of the ZUMA-7 phase 3 trial, which examined a total of 170 patients [13], axi-cel’s approved indications have been extended to relapsed or refractory Follicular Lymphoma (FL) after three or more lines of systemic therapy and to the second line treatment of DLBCL and High-Grade B-cell Lymphoma (HGBL). 

The safety of axi-cel has been investigated during pre-approval clinical trials, which showed clinically relevant Adverse Events (AEs) in most of the patients who received this CAR-T-cell [14]. Cytokine Release Syndrome (CRS), neurological toxicity, hypogammaglobulinemia, infection, and neutropenic fever are among the most-characteristic AEs reported with axi-cel. The term CRS designates a supraphysiologic systemic inflammatory response with heterogeneous clinical presentations and severity, caused by the activation of endogenous or infused lymphocytes induced by immune therapies [15]. The American Society for Transplantation and Cellular Therapy (ASTCT) classifies CRS severity using a five-grade system: Grade 1, in which fever ≥ 38 °C is the dominant symptom; Grade 2, in which fever is accompanied by hypoxia that can be managed with low-flow nasal cannulas; Grade 3, in which, besides fever, the patient also has hypoxia requiring high-flow nasal cannulas, a facemask, or a Venturi mask, and severe hypotension requiring a vasopressor drug with or without vasopressin; Grade 4, showing fever, hypotension requiring multiple vasopressor drugs, and hypoxia needing positive pressure ventilation; and, finally, Grade 5, which corresponds to patient death [15]. The release of cytokines and perforin/granzyme by CAR-T-cells upon activation by tumor antigens is considered the triggering factor for CRS since it causes the death of cancer cells with the consequent release of more cytokines and the further activation of T-lymphocytes and macrophages in a positive feedback loop [16,17]. Most of the clinical manifestations of CRS are the consequence of endothelial cell activation and damage by circulating cytokines; IL-6, acting through its soluble receptors, seems to have a dominant role in this process [17]. CRS treatment should be graded based on disease severity. According to the treatment guidelines of the American Society of Clinical Oncology (ASCO), only symptomatic and supportive measures are required for Grade 1, whereas tocilizumab, a humanized monoclonal antibody against IL-6 receptors, must be added in Grade 2; dexamethasone is given in the case of less-than-optimal response to tocilizumab [18]. Grade-3 patients should be treated with tocilizumab plus dexamethasone, whereas high-dose methylprednisolone in combination with tocilizumab is recommended for Grade-4 patients [18]. The most-typical form of neurotoxicity related to CAR-T-cells such as axi-cel and other forms of immune therapy is Immune-effector-Cell-Associated Neurotoxicity Syndrome (ICANS). It has been defined by the ASTCT as a neurological condition caused by the activation of endogenous or infused T-cells and characterized by symptoms that include aphasia, altered consciousness, impairment of the cognitive status, weakness, seizures, and cerebral edema [15]. Other neurological manifestations can be observed with axi-cel, such as agitation, ataxia, hallucinations, or tremor, but they are less specific and are not included in ICANS [15]. The pathogenesis of ICANS has been only partially clarified, but the key factor seems to be the permeabilization of the Blood–Brain Barrier (BBB) which is caused by circulating or locally released cytokines [16], inducing endothelial cell activation and damage and promoting the influx of CAR-T-cells in the cerebral parenchyma [19]. Four grades of ICANS severity have been identified by the ASTCT consensus based on clinical presentation and on a specific Immune-effector-Cell-associated Encephalopathy (ICE) score, which measures patient orientation, attention, and ability to follow commands, name objects, and write [15]. Grade 1 refers to patients with an ICE score of 7–9 who awaken spontaneously and have no seizures, motor symptoms, or cerebral edema. Grade-2 patients have an ICE score of 3–6 and still do not show seizures, motor symptoms, or cerebral edema, but they require voice stimuli to be awakened. In Grade-3 patients, whose ICE score is 0–2, tactile stimulation is required for awakening, seizures are present, and there is evidence of cerebral edema at neuroimaging. Finally, Grade 4-patients (ICE score of 0) cannot be awakened and have prolonged, serious seizures, diffuse weakness or paresis, and intense cerebral edema. The ASCO treatment guidelines recommend supportive measures for Grade-1 and -2 ICANS with the addition of dexamethasone in high-risk Grade-2 patients [18]. Corticosteroids (either dexamethasone or methylprednisolone) are the basis of the pharmacological treatment of Grade-3 and -4 patients. Whatever the ICANS grade is, tocilizumab should be added if the patient also has CRS. Since CD19 is expressed not only by neoplastic, but also by normal B-lymphocytes, axi-cel treatment can be complicated by B-cell aplasia and, consequently, hypogammaglobulinemia. This AE increases the risk of opportunistic infection, and if severe, it may require the intravenous administration of human immunoglobulins [18]. An additional factor that increases the risk of infections in patients treated with axi-cel is neutropenia, which can occur in isolation or in combination with anemia and/or thrombocytopenia or, more rarely, lymphocytopenia. The pathogenesis of CAR-T-cell-induced cytopenia is still uncertain, but it could involve off-target effects of released cytokines and the depletion of stromal cell-derived factor-1 [20,21]. Glucocorticoids can be used for the treatment of cytopenia of Grade 2 or higher, whereas hematological growth factors are recommended for Grade 4 patients [18]. Infections, more often presenting as neutropenic fever, are common in patients treated with axi-cel or other CAR-T-cells [22] and require the implementation of specific anti-infective drug therapies [18].

Since less than 400 patients were studied in pre-approval clinical trials, it is largely agreed that further information is required for a full characterization of axi-cel tolerability. Waiting for future clinical studies that will better characterize the risk–benefit profile of axi-cel, an additional and useful source of information could be represented by large pharmacovigilance databases, which record spontaneous reports of AEs, technically named Individual Case Safety Reports (ICSRs); specifically, ICSRs describe suspected AEs potentially related to drugs or vaccines that occur in real-world clinical use. In Europe, a centralized database known as EudraVigilance (EV) collects the ICSRs coming from Health Care Professionals (HCPs) and Non-Health Care Professionals (NHCPs), National Competent Authorities, post-approval studies, and the scientific literature [23]. The EV database, maintained by the EMA, represents a valuable tool to analyze safety data on already-approved drugs mostly coming from real-world utilization. In the present manuscript, we specifically analyzed all the spontaneous reports of AEs potentially related to axi-cel exposure in the EV database from 1 January 2018 to 31 December 2022.

## 2. Materials and Methods

### 2.1. Data Source

The data examined in the present paper were all obtained by interrogating the centralized European spontaneous reporting system, the EV database. This database stores the Individual Case Safety Reports (ICSRs) of suspected reactions related to the use of drugs (suspected ADRs) or vaccines (Adverse Events Following Immunization (AEFIs)). Since no causality assessment is included in report processing, all the suspected adverse drug reactions are better defined as AEs and will be referred to as such in the rest of the manuscript. EV includes two sections: the first one, the Post-Authorization Module (EVPM), is devoted to the collection of reports of AEs occurring with medicinal products already authorized by the EMA, whereas the second, the Clinical Trial Module (EVCTM), is for the collection of records of AEs occurring during clinical trials on products still in clinical experimentation. We only used data from the EVPM, which include both solicited and unsolicited records uploaded either by HCPs or by NHCPs. These data are freely accessible through a publicly available web portal (https://www.adrreports.eu/en/index.html, accessed on 30 March 2022) from which they can be retrieved either as ICSR forms or as a line listing.

Each EV ICSR summarizes the AEs that occurred during the treatment of a single specific patient with a drug or a vaccine that is suspected to be responsible for the recorded event. These reports are uploaded in the EV database using specific forms with multiple fields including data on the patient, his/her disease and therapy, and multiple details on the experienced AE (see Section 2.3 for more details) [24]. Multiple AEs can occur in a single patient and be included in a single ICSR. 

### 2.2. ICSRs Selection

We searched EV using as a keyword *Yescarta*, the brand name of axi-cel, the exposure of interest. All the ICSRs showing axi-cel as the suspected drug from 1 January 2018 (date of the first available ICSR in EV related to axi-cel) to 31 December 2022 were included in the analysis.

### 2.3. Descriptive Analysis

From each ICSR, we extracted general information both about the reporting source, the patient, and the suspected ADR. Specifically, we recorded whether the ICSR was uploaded by a HCP or a NHCP and whether the country of origin was from the European Economic Area (EEA) or non-European Economic Area (non-EEA). Retrieved patient information included demographic data (age and gender), clinical data (the disease that was reported in the ICSR as the clinical indication for axi-cel), and data about the other drugs taken by the patient and that had been included in the ICSR as concomitant or suspected and that we classified according to the ATC classification system. 

Retrieved data about the suspected ADR included the organ/system involved (which we classified using the System Organ Class (SOC) categories) and the clinical severity and outcome of the reaction, graded according to the categories included in the ICSR. Specifically, the severity can be rated as “serious” or “non-serious” depending on whether or not the AE was a congenital anomaly/birth defect, caused the death of the patient, was life-threatening, required or prolonged his/her hospitalization or determined persistent or significant disability/incapacity, or other clinically relevant conditions. By contrast, the outcome of the AE is graded in the following categories: “Recovered/Resolved”, “Recovering/Resolving”, “Recovered/Resolved with Sequelae”, “Not Recovered/Not Resolved”, “Fatal”, or “Unknown”. 

### 2.4. Disproportionality Analysis

To assess whether there was a significant difference in the frequency of reporting AEs included in the SOC “Nervous system disorders” with axi-cel and with the other approved CAR-T-cells, we performed a disproportionality analysis using Reporting Odds Ratios (RORs) with 95% confidence intervals. This statistical method uses contingency tables to compare the odds that a certain AE occurred with the test drug with the odds that it occurred with a comparator drug. The 95% Confidence Interval (CI) gives information on the precision of ROR assessment. 

### 2.5. Ethical Consideration

Since all the ICSRs are anonymized and do not include any information that could make the patient identifiable, no approval from the Ethics Review Board was deemed necessary.

## 3. Results

### 3.1. Demography and General Characteristics of the ICSRs with Axi-cel as Suspected Drug

A total of 2905 ICSRs with axi-cel as the suspected drug were found in the European Pharmacovigilance database EV from 1 January 2018 to 31 December 2022. Among these ICSRs, 1849 (63.6%) were from the non-EEA and the others from the EEA. Most ICSRs were reported by health care professionals (n = 2803; 96.5%). 

Overall, 1331 (45.8%) ICSRs were related to male patients and the described AEs mostly occurred in patients aged 18–64 years (42.1%) or older (aged 65–85 years) (22.6%) (Table 1).

DLBCL was the therapeutic indication more-frequently reported (n = 1369; 46.5% of reported indications) followed by B-cell lymphoma (n = 272; 9.2%), non-Hodgkin’s lymphoma (n = 111; 3.8%), PMBCL (n = 92; 3.1%), lymphoma (n = 85; 2.9%), and FL (n = 65; 2.2%). In 881 out of 2905 ICSRs, the information about the clinical indication for axi-cel prescription was missing and the respective field showed “Product used for unknown indication” (Table 2).

### 3.2. Reported Suspected Adverse Reactions

A total of 8982 AEs were reported in the 2,905 ICSRs that we examined, indicating that, in most cases, more than one AE occurred per patient. The majority of AEs were scored as serious (n = 7104; 79.1%), and the most-common seriousness criteria selected by the reporters were *Other Medically Important Condition* (n = 3621; 50.9%) and *Caused/Prolonged Hospitalization* (n = 2235; 31.4%), whereas the criteria *Resulting in Death*, *Life Threatening*, and *Disabling* were chosen in 12.3%, 4.7%, and 0.7% of the ICSRs, respectively (Table 3).

The outcome was not reported for 3730 AEs (41.5%), whereas 3194 (35.6%) AEs were classified as *Recovered/Resolved*, 852 (9.5%) *Not Recovered/Not Resolved*, 410 (4.6%) as *Recovering/Resolving*, and 31 (0.3%) as *Recovered with Sequelae*. Death was the reported outcome of 765 (8.5%) AEs, and in most cases, it was due either to disease progression, CRS, immune-mediated neurotoxicity, or hemophagocytic lymphohistiocytosis, a known AE associated with axi-cel therapy, which often occurs in the setting of CRS and is classified in the axi-cel Summary of Product Characteristics (SmPC) among immune system disorders as uncommon.

Figure 2 reports the results that we obtained when we classified axi-cel-related AEs according to the System Organ Class (SOC) categories. This analysis showed that “Nervous system disorders” (n = 2304; 25.6%) and “Immune system disorders” (n = 2078; 23.1%) represented more than half of the analyzed reported AEs. About 12.0% (n = 1105) of the AEs were classified in the SOC “General disorders and administration site conditions”, 7.2% (n = 652) in the SOC “Blood and lymphatic system”, and 5.8% (n = 517) in the SOC “Infections and infestations”. Each of the remaining SOCs accounted for less than 5% of axi-cel-related AEs.

Table 4 shows the detailed list of axi-cel-related AEs classified by SOCs. Neurotoxicity (n = 723; 31.4%) and Immune-effector-Cell-Associated Neurotoxicity Syndrome (ICANS) (n = 616; 26.7%) were the most-frequently reported AEs among those classified in the “Nervous system disorders” SOC, while cytokine release syndrome represented more than 90% of those belonging to the “Immune system disorders” SOC. Forty-five ICSRs included hemophagocytic lymphohistiocytosis as the AE, and in 34 of them (75.6%), it occurred together with CRS. 

*Pyrexia* accounted for 33% of all the disorders reported in in the SOC “General disorders and administration site conditions”. *Neutropenia*, *pancytopenia*, *thrombocytopenia*, and *cytopenia* represented 65% of the disorders classified in the SOC “Blood and lymphatic system”, whereas *sepsis*, *septic shock*, and *infection* were the most-common conditions in in the SOC “Infections and infestations”. 

Since the disorders included in the SOC “Nervous system disorders” represented the most-frequent group of AEs reported in axi-cel-related ICSRs and considering that neurological toxicity has been described also with other CAR-T-cells, we performed a disproportionality analysis to assess whether axi-cel was associated with a lower/higher probability of reporting ICSRs with nervous system AEs as compared with the other approved CAR-T-cells. To this aim, we calculated the ROR, with a 95% CI, of the SOC “Nervous system disorders” for axi-cel compared to tisa-cel, ide-cel, liso-cel, cilta-cel, and brexu-cel or to all of them as a single group. The results of this analysis showed that axi-cel was associated with a higher probability of reporting AEs belonging to the SOC “Nervous system disorders” in comparison with tisa-cel, ide-cel, liso-cel, and cilta-cel (ROR 3.03, 95% CI 2.70–3.39, *p* < 0.05; ROR 2.29, 95% CI, 1.68–3.19, *p* < 0.05; ROR 1.76 95% CI 1.17–2.74, *p* < 0.05; ROR 3.27 95% CI 1.92–5.99, *p* < 0.05, respectively) (Table 5). By contrast, the ROR of nervous system AEs was slightly, but non-significantly lower with axi-cel as compared with brexu-cel (ROR 0.91, CI 0.77–1.08, *p* = 0.26). 

### 3.3. Concomitant Medicinal Products

A total of 6152 concomitant medicinal products were found in the 2905 analyzed ICSRs. According to the 2nd Level Anatomical Therapeutic Chemical (ATC) classification, *Antineoplastic agents* (n = 1009; 16.4%) were the most-frequently reported group, followed by *Antibacterials for systemic use* (n = 598; 9.7%), *Ophthalmologic* (n = 502; 8.2%), *Analgesic* (n = 309; 5.0%), *Antiepileptics* (n = 306; 5.0%), *Drugs for acid related disorders* (n = 292; 4.7%), *Psycholeptics* (n = 230; 3.7%), *Blood substitutes and perfusion solutions* (n = 196; 3.2%), *Antithrombotic agents* (n = 179; 2.9%), and *Antifungals for dermatological use* (n = 171; 2.8%) (Table 6).

## 4. Discussion

During pre-approval clinical trials, axi-cel has been evaluated only on a limited number of patients, and therefore, additional information from the real-world use of this CAR-T-cell gene therapy is required for a full characterization of its tolerability profile. In this context, in the present paper, we summarized the data from the 2905 ICSRs with axi-cel as the suspected drug that were deposited in the EV database from 1 January 2018 to 31 December 2022. In agreement with previous studies, most of these ICSRs concerned males aged 18–64 years [25,26] who received axi-cel for therapeutic indications reported in the SmPC: DLBCL in more than 40% of cases, followed by B-cell lymphoma, non-Hodgkin’s lymphoma, PMBCL, and FL [14]. Our results showed that most of the reported AEs were serious. This finding, however, does not necessarily imply that axi-cel predominantly causes serious AEs, but could be the consequence of the fact that about 96.5% of the ICSRs were reported by HCPs, who, according to previous studies, seem to be especially prone to preferentially report more serious AEs [27,28,29]. Analyzing all reported AEs according to the SOC categories, we found that the most-frequently reported SOC was “Nervous system disorders” and that, in this SOC, ICANS was the AE most-often observed. As mentioned in the Introduction, this condition, which may cause neurological symptoms of varying severity from disorientation and inattentiveness to impaired consciousness, coma, and even death, is still one of the most-feared complications of CAR-T-cell therapy because of its clinical severity [30,31,32]. ICANS has been reported with all the approved CAR-T-cells with a prevalence that varies widely across studies ranging from 20 to 67% [10,33,34,35]. Very few comparative data are available to establish whether the probability of reporting ICANS or, more in general, Nervous system disorders differs significantly among different CAR-T-cells. Bonaldo and colleagues (2021) [25] and Fusaroli et al. (2022) [26], who analyzed both EV and the FDA Adverse Event Reporting System (FAERS) for axi-cel and tisa-cel ADRs, reported a higher frequency of reported neurotoxicity in axi-cel than with tisa-cel. In the present manuscript, we extended the comparison to other more-recently approved CAR-T-cells including cilta-cel, ide-cel, liso-cel, and brexu-cel. The results of our disproportionality analysis showed that the probability of reporting AEs belonging to the “Nervous system disorders” SOC was higher with axi-cel than with all the other CAR-T-cells considered, with the only exception of brexu-cel, which showed a frequency of reported neurological events comparable to axi-cel. These findings fit well with the prevalence of neurotoxicity reported in the European Public Assessment Report (EPAR) of each of these CAR-T-cells (65% for axi-cel, 57% for brexu-cel, 27% for tisa-cel and for liso-cel, 21% for cilta-cel (21%), and 18% for ide-cel) [36,37,38,39,40]. The reason why the prevalence of ICSRs reporting ICANS or, more in general, neurotoxicity is different among the different CAR-T-cells and higher with axi-cel and brexu-cel is unclear. However, some evidence from experiments performed with modified CAR-T-cells in animal models suggests that differences in the genetic constructs that are used for their preparation could be involved [32]. First, the difference in the antigen targeted by CAR-T-cells could be relevant. In fact, the most-neurotoxic of the examined CAR-T-cells were those targeting CD19, whereas the risk of neurotoxicity was markedly lower with cilta-cel and ide-cel, which are both directed against BCMA. As a matter of fact, by using scRNA-Seq, Parker et al. (2020) [41] showed that CD19 is expressed in a small fraction of mural cells, mainly pericytes, of the BBB, suggesting that CAR-T-cells targeting this antigen could also directly interact with these cells and cause their damage, ultimately leading to BBB permeabilization and neurotoxicity. Another factor that could help explain the difference in the neurotoxic potential of different CAR-T-cells is the costimulatory domain, which is bound to the CD3ζ intracellular portion of the chimeric antigen receptor in each specific CAR-T-cell type. In fact, both axi-cel and brexu-cel, the two CAR-T-cells with the highest neurotoxic potential, use CD28 for costimulation, whereas in tisa-cel, cilta-cel, and liso-cel, the other, less-neurotoxic CAR-T-cells directed against CD19, 4-1BB (CD137) is used. Intriguingly, CD28 and 4-1BB costimulatory domains confer different phenotypic and metabolic profiles to CAR-T-cells. More specifically, CAR-T-cells expressing the CD28 costimulatory domain have a T-cell effector phenotype, activate faster, mainly rely on glycolytic metabolism, and release higher amounts of cytokines, which could induce relevant toxicity. In contrast, the 4-1BB costimulatory domain promotes the development of memory T-cells, which mainly depend on mitochondrial respiration and release less cytokines for a longer time, being potentially less toxic than CD28-expressing CAR-T-cells [32,42,43].

The occurrence of immune disorders including CRS and hemophagocytic lymphohistiocytosis among the most-frequently reported AEs in axi-cel-related ICSRs was highly expected since they have been frequently observed in axi-cel preapproval clinical trials [10,44] and seem to be the consequence of cytokine release, which occurs initially because of T-cell activation with IL-6 release and later because of the death of leukemic cells and of the ensuing macrophage activation [16,45,46]. 

Likewise, the evidence that Infections and infestations ranked fifth among the most-frequently reported AEs in patients treated with axi-cel is in line with data from pre-approval clinical trials. Patients treated with axi-cel, similarly to patients treated with CAR-T-cells, are, indeed, at risk for infective complications because of multiple concomitant factors, which include hypogammaglobulinemia, cytopenia, and B-cell aplasia, and the lymphoablative treatment that is required before giving these innovative products [18]. 

Due to the severity of the underlying clinical conditions that led to the prescription of axi-cel, multiple concomitant medicinal products, mostly antineoplastic and anti-infective drugs, were reported in the majority of the ICSRs that we reviewed. Whether they contributed to the reported AEs or not cannot be established because of the limitations imposed by the type of data reported in the ICRS. It is worth mentioning, however, that some of the drugs commonly used in patients receiving CAR-T-cells, such as fludarabine, are known to be neurotoxic and to have a role in promoting the development of CRS [45,47]. The specific case of fludarabine has a special interest since it emphasizes the difficulties in diagnosing and classifying the neurological symptoms developing in patients treated with CAR-T-cells. In fact, Winter et al. (2021) [47] pinpointed several differences between fludarabine-related neurotoxicity and ICANS that could help the differential diagnosis between these two conditions, with the former consisting of a progressive neurological syndrome with visual impairment and the latter of transient symptoms dominated by speech disturbances.

The present study had several limitations, which all stemmed from the data having been obtained from a pharmacovigilance reporting database. This implies that no information is available on the actual size of the population of patients that received the treatment, of whom those who developed an ADR just represent a subset. 

## 5. Conclusions

In conclusion, we examined ICSRs from EV with axi-cel as the suspected drug to further characterize the safety profile of this CAR-T-cell. Our findings showed that the AEs most-frequently reported with this CAR-T-cell are nervous system and immunological disorders. Our disproportionality analysis provides some evidence suggesting that axi-cel could be associated with a higher neurotoxicity risk than other CAR-T-cells, but due to the intrinsic limitations of pharmacovigilance database investigations, well-designed comparative clinical trials will be required to confirm this hypothesis.

## Figures and Tables

**Figure 1 biomedicines-11-02162-f001:**
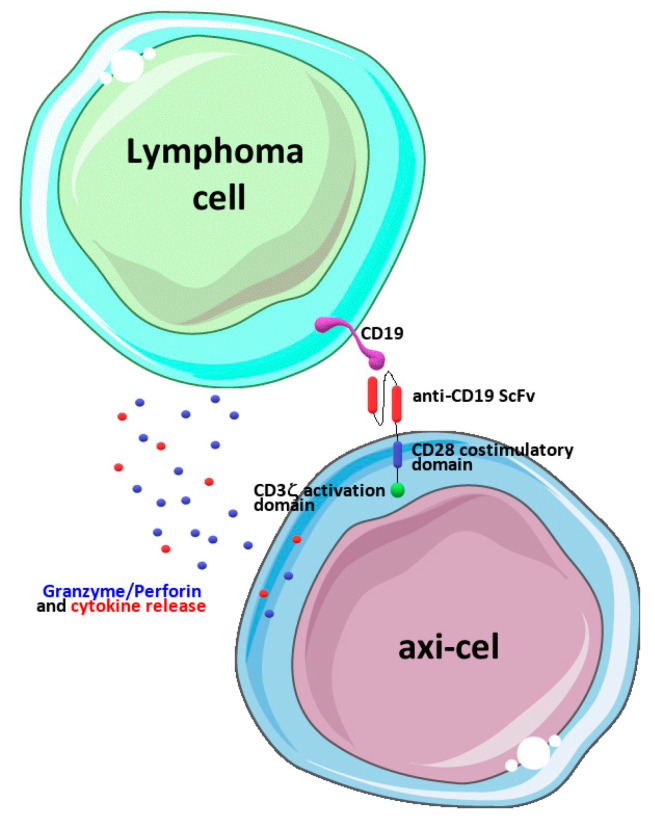
Schematic representation of axi-cel’s mechanism of action. Engineered CAR receptors including a CD28 costimulatory domain and a CD3ζ activation domain are engaged upon binding CD19 antigens on neoplastic lymphoma cells, and this leads to cell activation and the release of cytokines and granzymes/perforins, ultimately leading to the death of cancer cells. The drawing was prepared using image templates freely available from the smart Servier website (https://smart.servier.com/, accessed on 20 June 2023).

**Figure 2 biomedicines-11-02162-f002:**
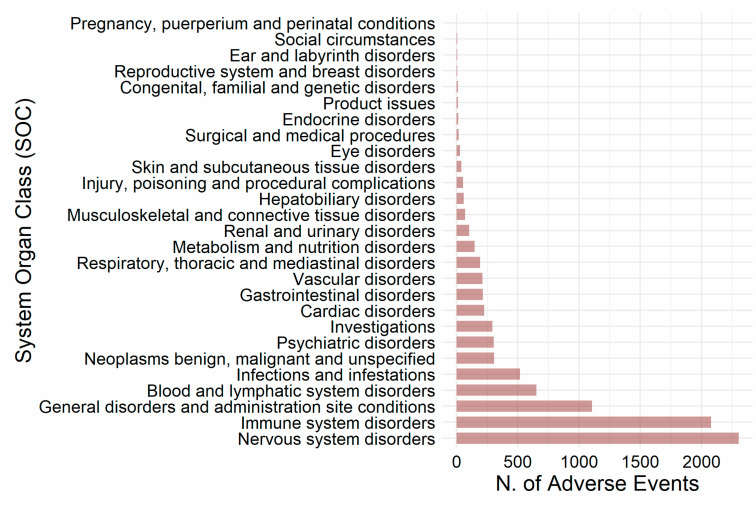
Distribution of axi-cel-related AEs according to the System Organ Class (SOC) classification.

**Table 1 biomedicines-11-02162-t001:** Distribution of axi-cel-related ICSRs by age, gender, reporter groups, and primary source country.

		ICSRs n (% of Total)
**Gender**	Female (%)	927 (31.9)
	Male (%)	1331 (45.8)
	Not Reported (%)	647 (22.3)
	**TOTAL**	**2905 (100.0)**
** *Age* **	12–17 Years (%)	1 (<0.1)
	18–64 Years (%)	1225 (42.1)
	65–85 Years	657 (22.6)
	More than 85 Years	6 (0.2)
	Not Specified	1016 (35.0)
	**TOTAL**	**2905 (100.0)**
** *Reporter Groups* **	Health Care Professional	2803 (96.5)
	Non-Health Care Professional	102 (3.5)
	**TOTAL**	**2905 (100.0)**
** *Primary Source Country* **	Non-EU Economic Area	1849 (63.6)
	EU Economic Area	1056 (36.4)
	**TOTAL**	**2905 (100.0)**

EU: European.

**Table 2 biomedicines-11-02162-t002:** Therapeutic indications reported in axi-cel-related ICSRs.

Therapeutic Indication	ICSRs n (% of Total)
Diffuse large B-cell lymphoma	1369 (46.5)
B-cell lymphoma	272 (9.2)
Non-Hodgkin’s lymphoma	111 (3.8)
Primary mediastinal large B-cell lymphoma	92 (3.1)
Lymphoma	85 (2.9)
Follicular lymphoma	65 (2.2)
TOTAL	2945 (100.0) *

The table summarizes the data on the frequency distribution prevalence of the different therapeutic indications reported for axi-cel in each of the selected ICSRs. Therapeutic indications with a prevalence lower than 2% are not included in the table. * The discrepancy between the total number of therapeutic indications (n = 2945) and the total number of retrieved ICSRs (n = 2905) can be explained considering that more than one therapeutic indication was reported in some of the retrieved ICSRs.

**Table 3 biomedicines-11-02162-t003:** Seriousness and outcome of AEs related to axi-cel.

		AEs n (%)
**Seriousness**	Serious (%)	7104 (79.1)
	Not Serious (%)	1878 (20.9)
	TOTAL	8982 (100.0%)
**Outcome**	Recovered/Resolved	3194 (35.6)
	Recovering/Resolving	410 (4.6)
	Recovered with Sequelae	31 (0.3)
	Not Recovered/Not Resolved	852 (9.5)
	Fatal	765 (8.5)
	Unknown	3730 (41.5)
	TOTAL	8982 (100.0%)

For each seriousness category, as well as for each outcome group, the number of ICSR AEs for which this information was available and the relative percentage with respect to the total of AEs are reported.

**Table 4 biomedicines-11-02162-t004:** Axi-cel-related AEs classified by SOCs.

	AEs n (%)
**Nervous system disorders**	**2304 (100.0)**
Neurotoxicity	723 (31.4)
ICANS	616 (26.7)
Encephalopathy	154 (6.7)
Tremor	105 (4.6)
Aphasia	92 (4.0)
Headache	84 (3.6)
Somnolence	53 (2.3)
Dysgraphia	41 (1.8)
Seizure	34 (1.5)
Memory impairment	29 (1.3)
Depressed level of consciousness	23 (1.0)
**Immune system disorders**	**2078 (100.0)**
Cytokine release syndrome	1929 (92.8)
Hypogammaglobulinemia	54 (2.6)
Hemophagocytic lymphohistiocytosis	45 (2.2)
Cytokine storm	30 (1.4)
**General disorders and administration site conditions**	**1105 (100.0)**
Pyrexia	368 (33.3)
Disease progression	155 (14.0)
Death	141 (12.8)
Fatigue	116 (10.5)
Malaise	36 (3.3)
Drug ineffective	36 (3.3)
Chills	36 (3.3)
Disease recurrence	20 (1.8)
Multiple organ dysfunction syndrome	19 (1.7)
Asthenia	18 (1.6)
Gait disturbance	13 (1.2)
Hyperthermia	12 (1.1)
**Blood and lymphatic system disorders**	**652 (100.0)**
Neutropenia	176 (27.0)
Pancytopenia	97 (14.9)
Thrombocytopenia	87 (13.3)
Cytopenia	70 (10.7)
Febrile neutropenia	44 (6.7)
Bone marrow failure	34 (5.2)
Anemia	30 (4.6)
Leukopenia	17 (2.6)
Myelosuppression	14 (2.1)
Disseminated intravascular coagulation	14 (2.1)
B-cell aplasia	14 (2.1)
Coagulopathy	13 (2.0)
Lymphopenia	7 (1.1)
Agranulocytosis	7 (1.1)
**Infections and infestations**	**517 (100.0)**
Sepsis	39 (7.5)
Septic shock	36 (7.0)
Infection	35 (6.8)
Pneumonia	32 (6.2)
COVID-19	24 (4.6)
Bacteremia	17 (3.3)
Cytomegalovirus infection reactivation	11 (2.1)
Clostridium difficile colitis	10 (1.9)
Aspergillus infection	10 (1.9)
Urinary tract infection	9 (1.7)
Systemic candida	9 (1.7)
Clostridium difficile infection	9 (1.7)
Systemic mycosis	7 (1.4)
Bronchopulmonary aspergillosis	7 (1.4)
Pneumonia bacterial	6 (1.2)
Human herpesvirus 6 encephalitis	6 (1.2)
BK virus infection	6 (1.2)
Staphylococcal infection	5 (1.0)
Pneumonia fungal	5 (1.0)

For each AE belonging to a specific SOC, the total number of the ICSRs reporting it and the percentage that this AE represents with respect to the total number of all the AEs in the same SOC are reported. AEs with a prevalence lower than 1% are not included in the table.

**Table 5 biomedicines-11-02162-t005:** Disproportionality analysis of the reporting in ICSRs of events related to the SOC “Nervous system disorders” with axi-cel and the other approved CAR-T-cells.

Comparisons	ROR	95% CI	*p*-Value
axi-cel vs. all	2.34	2.13–2.58	<<0.05
axi-cel vs. tisa-cel	3.03	2.70–3.39	<<0.05
axi-cel vs. brexu-cel	0.91	0.77–1.08	0.26
axi-cel vs. ide-cel	2.29	1.68–3.19	<<0.05
axi-cel vs. liso-cel	1.76	1.17–2.74	<0.05
axi-cel vs. cilta-cel	3.27	1.92–5.99	<<0.05

The table shows the ROR with respective 95% CI and *p*-values of ICSRs with ADRs belonging to the SOC Nervous system disorders to compare the probability of reporting these AEs for the comparisons of axi-cel versus all approved CAR-T therapies, axi-cel versus tisa-cel, axi-cel versus brexu-cel, axi-cel versus ide-cel, axi-cel versus liso-cel, and axi-cel versus cilta-cel. CI: Confidence Interval.

**Table 6 biomedicines-11-02162-t006:** Classes of concomitant drugs reported with axi-cel.

2nd Level ATC Drug Classes		n (%)
Antineoplastic agents	(L01)	1009 (16.4)
Antibacterials for systemic use	(J01)	598 (9.7)
Ophthalmic drugs	(S01)	502 (8.2)
Analgesic	(N02)	309 (5.0)
Antiepileptic drugs	(N03)	306 (5.0)
Drugs for acid related disorders	(A02)	292 (4.7)
Psycholeptics	(N05)	230 (3.7)
Blood substitutes and perfusion solutions	(B05)	196 (3.2)
Antithrombotic agents	(B01)	179 (2.9)
Antifungals for dermatological use	(D01)	171 (2.8)
Drugs for constipation	(A06)	161 (2.6)
Antivirals for systemic use	(J05)	154 (2.5)
Musculo-skeletal system	(M04)	147 (2.4)
Antiemetics and antinauseants	(A04)	128 (2.1)
Antihistamines for systemic use	(R06)	121 (2.0)
Corticosteroids, dermatological preparations	(D07)	117 (1.9)
Antimycotics for systemic use	(J02)	106 (1.7)
Beta blocking agents	(C07)	99 (1.6)
Psychoanaleptics	(N06)	98 (1.6)
Drugs used in diabetes	(A10)	95 (1.5)
Immunosuppressants	(L04)	92 (1.5)
Mineral supplements	(A12)	88 (1.4)
Diuretics	(C03)	70 (1.1)
Antineoplastic agents	(L01)	1009 (16.4)

Concomitant drugs reported with axi-cel were classified as 2nd Level Anatomical Therapeutic Chemical (ATC) groups. For each drug group, the table shows both the number of reports and the percentage of all concomitant drugs (n = 6152) that it represents. Concomitant drugs accounting for less than 1% of all concomitant drug prescription are not included in the table.

## Data Availability

No new data were created to prepare this article, which was entirely based on the review of publicly available ICSRs retrieved from the EudraVigilance database.

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
