# Peer review of "A Pharmacovigilance Study on the Safety of Axicabtagene Ciloleucel Based on Spontaneous Reports from the EudraVigilance Database"

_biomedicines, 2023, doi:10.3390/biomedicines11082162_

Round 1
Reviewer 1 Report
This work focuses on the secondary effects of Axicabtagene Ciloleucel (Yescarta).
According to the FDA, the indication of this drug is the treatment of adult patients with large B-cell lymphoma that is refractory to first-line chemoimmunotherapy or that relapses within 12 months of first-line chemoimmunotherapy. It is also used in follicular lymphoma, relapsed or refractory.
The manuscript is well written, it is easy to understand, and have enough tables. It focuses only on Yescarta, but Table 5 compares with other similar drugs, so I got some confusion at the beginning.
Specific comments:
1) Line 76, regarding "BCMA", could you please use "TNFRSF17" as well?
2) In the introduction, if this manuscript focuses on Yescarta, why not providing more information about this drug?
** The two important toxicities:
Cytokine release syndrome (CRS). Grade 1, 2, 3, and 4. Treatment, tocilizumab and corticosteroids.
Neurologic toxicity. Grades 1, 2, 3, and 4; and management, dexamethasone, methylprednisolone.
other:
hypogammaglobulinemia
infection
netropenic fever
** Is it possible to show in a figure the mechanism of action of this drug?
** Is it possible to show in figure and/or text the pathological backgroud (mechanism) of the secondary effects?
** Why not making a table with all adverse reactions and their frequency?
cardiovascular, dematologic, endocrine, gastrointestinal, hepatic, etc...
3) Table 1 shows the distribution of individual case safety reports (ICSRs) of Yescarta. The total were 2,905 ICSRs, but, are these number of patients with adverse reactions, or one patient can have more than 1 adverse reaction reported?
4) In Table 2 the terms "B-cell lymphoma, non-Hodgkin lymphoma, and lymphoma" are very generic. I understand that the database may not provide more specific details. Is this the case? DLBCL is a B-cell lymphoma of the non-hodgkin lymphoma subtype.....
5) What is the difference between ICSRs and ADRs?ICSR = patient; ADR = 1 reaction ?
6) There are many abbreviations, an abbreviation list at the end of the discussion would help the readers.
Reviewer 2 Report
The authors investigated safety profiles of CAR-T cell using a database and revealed that the adverse events most frequently reported with the CAR-T cell were nervous system and immunological disorders. This work is significant in the clinical settings, whereas the methodology is wrong. The authors focused on axi-cel but data included other CAR-T cells. On the other hand, the authors compared axi-cel with others. A flow diagram for the selection of eligible data should be created, and data extraction need to be rethink. Based on the revision, the authors should discuss axi-cel, especially significant differences on nervous system disorders and mechanisms of the differences.
Round 2
Reviewer 2 Report
The authors revised appropriately. No further correction is necessary.